

# Effect of nitrogen fertilizer on seed yield and quality of *Kengyilia melanthera* (Triticeae, Poaceae)

Shuai Yuan[1,*], Yao Ling[1,*], Yi Xiong[1], Chenglin Zhang[1], Lina Sha[1], Minghong You[2], Xiong Lei[2], Shiqie Bai[2] and Xiao Ma[1]

[1] Sichuan Agricultural University, Chengdu, China
[2] Sichuan Academy of Grassland Sciences, Chengdu, China
* These authors contributed equally to this work.

Corresponding authors
Shiqie Bai, baiforage@163.com
Xiao Ma, maroar@126.com

## ABSTRACT

Widely distributed in the alpine sandy grassland in east Qinghai-Tibet Plateau (QTP), *Kengyilia melanthera* is considered as an ideal pioneer grass for the restoration of degraded and desertification grassland in the region. Under the special ecological and climatic conditions in the northwest Sichuan plateau located in east QTP, it is of great significance to optimize the amount of nitrogen fertilizer for the seed production of this species. The impact of nitrogen (N) fertilizer application on seed yield and quality of *K. melanthera* 'Aba', the only domesticated variety in the *Kengyilia* genus of Poaceae, was investigated based on two-year field experiments in the northwestern Sichuan plateau. The results showed that with the increase of N fertilizer application, the number of tillers, number of fertile tillers, 1,000-seed weight and seed yield of this species increased likewise. The optimum N fertilizer rate deduced in the present study was 180 kg·hm$^{-2}$, where the number of fertile tillers 1,000-seed weight and seed yield reached the peak values. Interestingly, the standard germination rate, germination energy, accelerated aging germination rate, dehydrogenase and acid phosphatase activity of seeds were not affected by the increasing the input of N fertilizer. The comprehensive evaluation of membership function showed that the optimal N fertilizer treatment was 180 kg·hm$^{-2}$ both for 2016 and 2017. This study provided a certain practical suggestion for the improvement of seed production of *K. melanthera* in the northwest Sichuan plateau.

## INTRODUCTION

The Qinghai-Tibet Plateau (QTP), the so-called third pole of the world, is an important eco-region. The grassland ecosystem in the QTP is undergoing extensive and longtime deterioration and even desertification, due to both human and natural factors (*Lu et al., 2017*; *Wu, Yu & Jin, 2017*). Since the beginning of the 21st century, the Chinese government has implemented large-scale ecological restoration projects to conserve

grasslands, mitigate degradation, and combat desertification (*Ma, Diao & Deng, 2019*). However, the return of a desertified grassland to a more natural state by incorporation of phytorestoration technology has been limited chiefly due to the lack of suitable grass species and sufficient seed supply (*Yu et al., 2019*; *Atinkut et al., 2020*). Notably, *Kengyilia melanthera*, a perennial grass species, is widely distributed the alpine sandy grassland in east QTP, lives in habitats at an altitude of 3,300–4,800 m (*Yang & Yan, 1992*). It is an important ecological barrier against desertification because of its strong adaptation to arid/ semiarid climates and barren sandy soil (*Xiao et al., 2008*). To date, a unique authorized *Kengyilia* variety in China bred by Sichuan Academy of Grassland Science, namely *K. melanthera* 'Aba', has been approved by the National Herb Variety Approval Committee of China in 2009 (*Xiao et al., 2011*).

K. melanthera is one of the few grass species that can produce seed on a large scale in east QTP. Nevertheless, the high exigency of seed utilization for ecological recovery of pasture builds demand for the more sufficient and high-quality seeds of *K. melanthera*. This results mainly from low productivity of native seed fields and the subsequent low efficiency of ecological restoration. Heredity and external conditions determine seed yield, and fertilization is one of the most effective ways to improve seed yield in extrinsic conditions. Usually, nitrogen (N) fertilizers are considered more effective for increasing crop seed yield than phosphate or potassium fertilizers (*Amanullah et al., 2016*; *Li et al., 2018a*). In QTP, animal manure was often collected for production of commercial N fertilizer while its N matter were relatively low. Therefore, addition of N fertilizer and screening out optimal dose to ensure the growth and development of plants is imperative (*Zhang et al., 2016*; *Quang et al., 2004*). *Wan, Wu & Liao (2021)*'s study on winter wheat with amounts of N fertilizer application of 0–360 kg·hm$^{-2}$ demonstrated that the optimal N application rate (240 kg·hm$^{-2}$) could not only increase the number of tillers, seed number per spike, and seed yield, but also could accelerate their growth process. *Xie et al. (2010)*'s study that examined the impact of amounts of N fertilizer application (0–330 kg·hm$^{-2}$) on wild *Elymus nutans* in Tibet found that the number of tillers, number of fertile tillers, and seed yield reached the peak value under 250 kg·hm$^{-2}$ and was significantly different from the other treatments. The genetic improvement and breeding of *K. melanthera*, as yet, mainly relies on mixed selection or recurrent selection of local wild germplasm. This led to a relatively low genetic gain especially on seed traits. As a result, it is an economical and practical option to increase the seed production potential of the of *K. melanthera* through cultivating measures such as nitrogen application. Moreover, we also found that harvesting seeds of *K. melanthera* at approximately 30 days after peak anthesis is considered as an optimal time point, which also has the lowest seed shattering occurrence.

The high demand of *K. melanthera* seeds for the recovery of desertified grassland in the QTP has been a long time, but not yet founded effective localized technology scheme. In order to exploit high-efficiency and inexpensive technical measures for seed production of *K. melanthera* 'Aba', a 2-year investigation was carried out to assess the impact of N fertilization on the seed yield and quality in the northwest Sichuan plateau, located in east of QTP, China.

# MATERIALS AND METHODS

## Experimental sites

The experiment was conducted for two consecutive years, between 2015 and 2017. The field is located in Hongyuan county, Aba Autonomous Prefecture, Sichuan province, a semi-humid region of the eastern QTP of China (31°79′35″N, 102°55′47″E, 3,480 m a.s.l.). Before sowing, the grasses and weeds were eliminated in all plots by hand weeding and herbicide spray, then the field was ploughed twice. The soil of experimental land was a subalpine meadow soil in nature, and chemical properties were provided in Table S1. In the 0–30 cm soil layer (pH 6.02), the concentration of organic matter, total nitrogen, total phosphorus, total potassium, available nitrogen, phosphorus, and potassium in the soil was 13.8, 1.32, 0.86, 0.68, 87.47, 62.55, and 227.46 mg·kg$^{-1}$, respectively.

## Experimental management

Seeds of *K. melanthera* (variety named 'Aba') was supplied by the Sichuan Academy of Grassland Sciences, and was harvested in the most recent production year before the field plot experiment. Its 1,000-seed weight and germination rate were 4.97 g and 80%, respectively. Generally, during the sowing year of (2015), the vast majority of *K. melanthera* plants growth, development, flowering, and seed setting are markedly reduced by the continental plateau's cold temperate monsoon climate. Hence, the seed yield and quality data in this study were recorded beginning from the second year (2016).

## Experimental design

The fertilizer treatments (urea fertilizer, 46% N) were applied to the experimental plots during the elongation stage, in mid June of 2016 and 2017. This trial is adopted as was a randomized complete block design for eight sample plots with different amounts of nitrogen fertilization ($N_1$ – 0 kg·hm$^{-2}$, $N_2$ – 60 kg·hm$^{-2}$, $N_3$ – 90 kg·hm$^{-2}$, $N_4$ – 120 kg·hm$^{-2}$, $N_5$ – 150 kg·hm$^{-2}$, $N_6$ – 180 kg·hm$^{-2}$, $N_7$ – 210 kg·hm$^{-2}$, $N_8$ – 240 kg·hm$^{-2}$). Each N fertilizer treatment had four replications' plots and a total of 32 plots. Sowing was performed by the drill method on May 20, 2015, with a seed density of 22.5 kg·hm$^{-2}$, sowing depth of 1–2 cm, and 50-cm row spacing. Experimental units consisted of plots with an area of 20 m$^2$ (5 m × 4 m) and 8 rows, and seeds were not sown within 50 cm of the boundary of the plot in order to reduce the marginal effect.

## Evaluations

### Seed yield components

The number of tillers (NTs), number of fertile tillers (NFTs), number of spikes (NSPs), number of florets (NFLs), number of fertile florets (NFFLs), and 1,000-seed weight (TSW) were measured to evaluate the seed yield components. At full-blooming stage (the proportion of plants that flowered reached 50%), a row was randomly selected from each sample plot for measurement of the NTs and NFTs, and four replicate measurement rows were performed within each plot. Here the length of selected row is 1 m and not on border rows of plot. In addition, the different indices cannot be sampled from the same plant. The NSPs, NFLs, and NFFLs were counted based on ten randomly sampled tillers in
full-blooming stage from each plot. The seeds from each plot were harvested from the 30th day after peak anthesis. 1,000 sun-dried seeds were randomly selected to determine the TSW, four replications were carried out.

### Seed yield

The seed yield indices could be represented by the harvested seed yield (HSY), potential seed yield (PSY), and presentation seed yield (PRSY). At the 30th day after peak anthesis, for each measurement, four row-segments with the length of 1 m were randomly selected from each plot, and all its seeds were harvested. After these seeds were threshed, cleaned, and naturally sun-dried to a seed moisture content of about 150 g·kg$^{-1}$, and the HSY of each N treatment could be obtained. The PSY and PRSY per unit area were calculated with the following formulas:

$$PSY = NFTs/m^2 \times NSPs/NFTs \times NFLs/NSPs \times \text{Average seed weight}$$

$$PRSY = NFTs/m^2 \times NSPs/NFTs \times NFFLs/NSPs \times \text{Average seed weight}$$

### Seed quality

To reduce or avoid dormancy and dormancy and after-ripening of freshly harvested seeds, seed quality trait analysis was tested approximately two months following harvest. The seed quality indices were based on standard germination rate (SGR), germination energy (GE), accelerated aging germination rate (AAGR), dehydrogenase (DE) and acid phosphoesterase (APH) activity. Because some weed seeds might be mistakenly harvested when *K. melanthera* seeds were harvested, so these seeds should be removed, the remaining seeds are called pure seeds. We randomly selected 50 pure seeds under different N treatments respectively and placed them into a Petri dish with two layers of filter paper on the bottom, adding a suitable amount of distilled water (preferably submerging seeds), and then moved them to cultivate in a plant growth chamber at a day temperature of 25 °C (16 h) and night temperature of 16 °C (8 h). Finally, we determined the SGR and GE of seeds on the 4th and 7th days of seed germination, respectively. For the seed-aging treatment, 250 pure seeds were randomly selected and embedded into an aging tank kept at 41 °C (72 h). The AAGR was measured after aging treatment, measurements from SGR and this method were identical. The DE and APH activity of seeds were determined by the triphenyl tetrazolium chloride (TTC) (*Mao et al., 2001*) and the sodium p-nitrophenol phosphate method (*Stephen, Daniel & William, 1989*), respectively. The assays described above were repeated four times.

## Statistics analysis

The statistical analysis was conducted by one-way and two-way analysis of variance (ANOVA) using IBM SPSS Statistics 20.0 software. The two-way analysis of ANOVA with a general linear model at $P < 0.05$ was applied to determine the combined influence of N fertilizer and trial years on seed yield components, seed yield, and quality. Bonferroni adjustment was applied after multiple hypothesis testing at the $P < 0.05$ probability level.

**Table 1 Analysis of variance of the influence of N fertilizer on *K. melanthera* seed yield components.**

| Index | Source of variance | Df | F-ratio | P-value |
|---|---|---|---|---|
| NTs /m$^2$ | Year (Y) | 1 | 5.313 | 0.026 |
| | N treatment (N) | 7 | 36.021 | 0.000 |
| | Interaction Y×N | 7 | 4.547 | 0.001 |
| NFTs /m$^2$ | Year (Y) | 1 | 7.790 | 0.008 |
| | N treatment (N) | 7 | 28.417 | 0.000 |
| | Interaction Y×N | 7 | 2.903 | 0.018 |
| NSPs per fertile tillers | Year (Y) | 1 | 0.011 | 0.916 |
| | N treatment (N) | 7 | 1.997 | 0.059 |
| | Interaction Y×N | 7 | 0.293 | 0.956 |
| NFLs per spike | Year (Y) | 1 | 0.799 | 0.373 |
| | N treatment (N) | 7 | 3.802 | 0.071 |
| | Interaction Y×N | 7 | 0.082 | 0.099 |
| NFFLs per spike | Year (Y) | 1 | 0.848 | 0.361 |
| | N treatment (N) | 7 | 104.774 | 0.000 |
| | Interaction Y×N | 7 | 0.259 | 0.966 |
| TSW/g | Year (Y) | 1 | 0.296 | 0.588 |
| | N treatment (N) | 7 | 273.820 | 0.000 |
| | Interaction Y×N | 7 | 5.615 | 0.000 |

**Note:**
NTs, number of tillers; NFTs, number of fertile tillers; NSPs, number of spikes; NFLs, number of florets; NFFLs, number of fertile florets; TSW, 1,000-seed weight; Df, degrees of freedom.

TBtools software is used to draw heat maps and other graphs were visualized using GraphPad Prism 8 procedure.

The membership function method was used to explore the response of seed yield components, yield, and quality to N fertilizer, and to determine the best fertilizer application rate for *K. melanthera*. The calculation formula for the forward membership function was $y = (x_a - x_{min})/(x_{max} - x_{min})$, and the negative membership function was $y = 1 - (x_a - x_{min})/(x_{max} - x_{min})$, where the $x_a$ is the value of certain index, and the $x_{max}$ and $x_{min}$ represent the maximum and minimum values in the same index of seed traits.

# RESULTS

## The influence of N fertilizer on seed yield components

Because of a typical plateau continental climate, plant growth and flowering of *K. melanthera* were restrained in the sowing year (2015). Therefore, we collected the seed yield-related and quality-related data from 2016 and 2017. N fertilizer had a significant influence ($P < 0.05$) on some yield components of seed including number of tillers (NTs), number of fertile tillers (NFTs), number of fertile florets (NFFLs) per spike, and 1,000-seed weight (TSW, Table 1). And non-significant difference was observed in all indices detected in 2016 and 2017.

Compared to 0 kg·hm$^{-2}$ (N$_1$, check), the NTs and NFTs in both trial years were significantly increased when 90–240 kg·hm$^{-2}$ (N$_3$–N$_8$) of N fertilizer was applied (Fig. 1A).

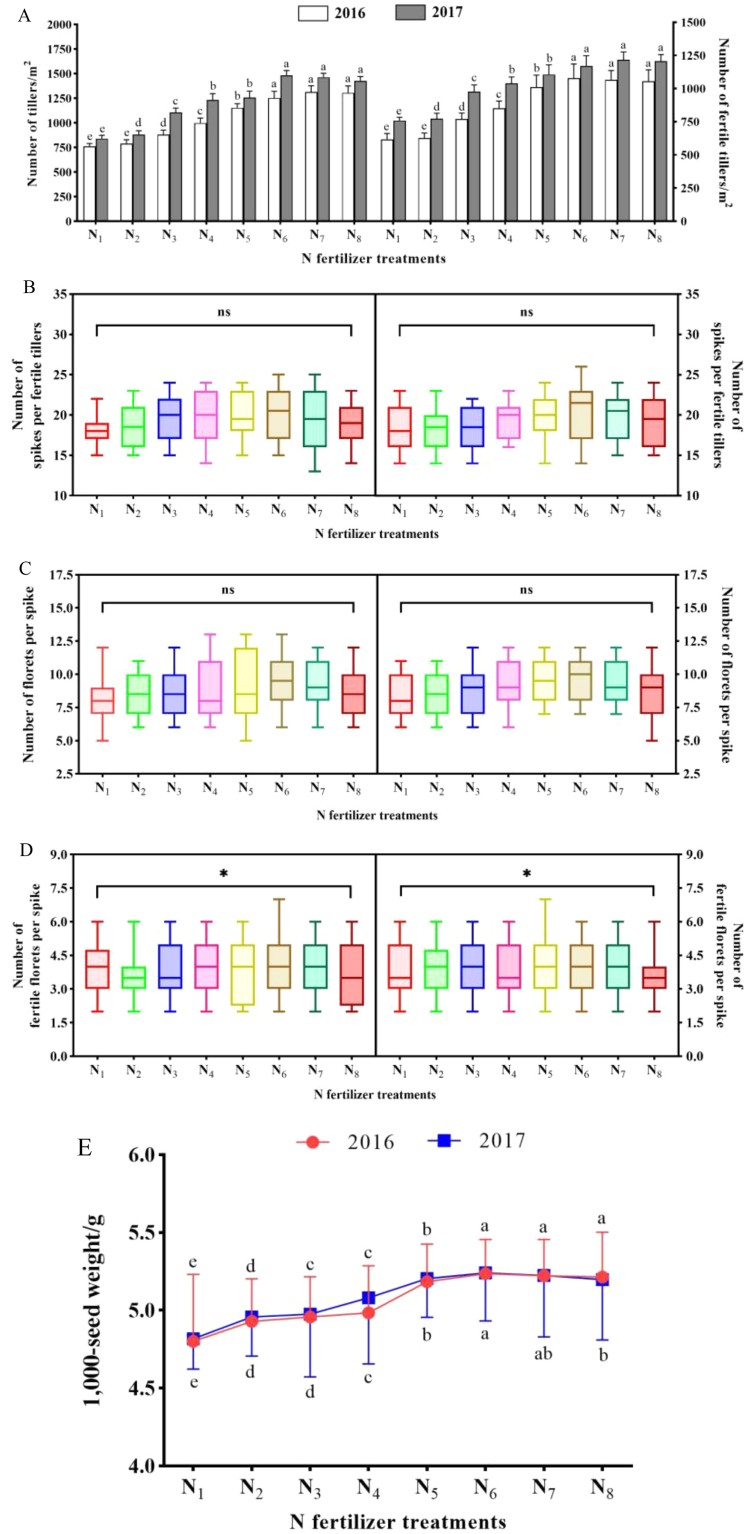

**Figure 1 The seed yield components of *K. melanthera* as influenced by eight different rates of N fertilizer application during the trial years 2016–2017.** (A) Number of tillers (mean ± standard error, *n* = 4), and fertile tillers (*n* = 4); (B) Number of spikes per fertile tiller (*n* = 10); (C) Number of florets per spike (*n* = 10); (D) Number of fertile florets per spike (*n* = 10); (E) 1,000-seed weight (*n* = 4).

**Figure 1 (continued)**
For box plots, top and bottom of box represent 75th and 25th percentiles, respectively. Different lower-case letters above the column and curve indicate significant differences under different N fertilizer treatments in each trial year (Bonferroni, $P < 0.05$). Vertical bar represents the standard error of mean. Statistical significance was determined as * $P < 0.05$, ns was $P > 0.05$. Note: $N_1$ – N fertilizer 0 kg·hm$^{-2}$ (Control); $N_2$ – N fertilizer 60 kg·hm$^{-2}$; $N_3$ – N fertilizer 90 kg·hm$^{-2}$; $N_4$ – N fertilizer 120 kg·hm$^{-2}$; $N_5$ – N fertilizer 150 kg·hm$^{-2}$; $N_6$ – N fertilizer 180 kg·hm$^{-2}$; $N_7$ – N fertilizer 210 kg·hm$^{-2}$; $N_8$ – N fertilizer 240 kg·hm$^{-2}$.                            

And they reached a maximum value under 180 kg·hm$^{-2}$ ($N_6$) or 210 kg·hm$^{-2}$ ($N_7$). The results showed that application of N fertilizer had a non-significant effect on number of spikes (NSPs) per fertile tillers and number of florets (NFLs) per spike in both trial years ($P > 0.05$, Figs. 1B and 1C). Compared to $N_1$, the NFFLs per spike of $N_2$ to $N_8$ were significantly different, and the peak values were both observed under $N_6$ treatment: 4.1 in 2016 and 4.2 in 2017 (Fig. 1D). The change trend of TSW is similar to that of NTs and NFTs, revealing that application of N fertilizer had a remarkable impact on TSW (Fig. 1E). It is worth noting that we observed a marginal decline ($P > 0.05$) in NTs, NFTs, NFFLs per spike, and TSW when the application amount of N fertilizer was over $N_7$ in 2016 and 2017, which may be due to the negative effect of excess N fertilizer.

## The influence of N fertilizer on seed yield and quality
The doses of N fertilizer application had a significant influence on the seed yield (Table 2, $P < 0.05$), but a non-significant effect in improving seed quality (standard germination rate (SGR), germination energy (GE), accelerated aging germination rate (AAGR), dehydrogenase (DE) and acid phosphoesterase (APH) activity, $P > 0.05$). With increasing rates of N fertilizer application rate, three seed yield indices (harvested seed yield (HSY), potential seed yield (PSY), and presentation seed yield (PRSY)) showed a tendency of first increasing and then decreasing (Figs. 2A and 2B), and reached their peak values under $N_6$ treatment in both 2016 and 2017. What is noteworthy is that excessive rates of N fertilizer application ($N_7$–$N_8$) had a non-significant suppression or improvement on seed quality, however, was accompanied by a reduction in three seed yield indices (Fig. 2, Table 3). During two-year independent field trials, we found that seed yield in 2017 was higher than that in 2016 under the same N fertilizer treatment, but the difference was non-significant.

## The relationship of N application and seed yield or seed components
N application had a significant influence on seed yield of *K. melanthera*, which could be explained by the increase of seed quantity and weight under the N effect (Table 4). By means of path analysis, we found that seed yield was significantly correlated with number of fertile tillers/m$^2$ (NFTs/m$^2$, r = 0.944, $P < 0.05$), 1,000-seed weight (TSW, r = 0.912, $P < 0.05$), and number of fertile florets (NFFLs) per spike (r = 0.899, $P < 0.05$), but not with number of spikes (NSPs) per fertile tiller (r = 0.806, $P > 0.05$) and number of florets (NFLs) per spike (r = 0.818, $P > 0.05$). Also, seed yield (r = 0.909, $P < 0.05$), NFTs/m$^2$ (r = 0.950, $P < 0.05$) and TSW (r = 0.921, $P < 0.05$) were observably associated with N treatment. It was therefore evident that N application was one of the most effective ways to enhance seed yield of *K. melanthera*.

**Table 2 Analysis of variance of the influence of N fertilizer on harvested seed yield and seed quality.**

| Index | Source of variance | Df | F-ratio | P-value |
|---|---|---|---|---|
| HSY/kg·hm$^{-2}$ | Year (Y) | 1 | 0.291 | 0.113 |
| | N treatment (N) | 7 | 145.194 | 0.000 |
| | Interaction Y×N | 7 | 0.283 | 0.958 |
| SGR/% | Year (Y) | 1 | 0.073 | 0.074 |
| | N treatment (N) | 7 | 16.779 | 0.983 |
| | Interaction Y×N | 7 | 0.607 | 0.747 |
| GE/% | Year (Y) | 1 | 0.049 | 0.850 |
| | N treatment (N) | 7 | 18.823 | 0.920 |
| | Interaction Y×N | 7 | 0.404 | 0.895 |
| AAGR/% | Year (Y) | 1 | 0.058 | 0.792 |
| | N treatment (N) | 7 | 17.348 | 0.772 |
| | Interaction Y×N | 7 | 0.265 | 0.821 |
| DE activity/µg·mL$^{-1}$ | Year (Y) | 1 | 1.654 | 0.203 |
| | N treatment (N) | 7 | 2.168 | 0.061 |
| | Interaction Y×N | 7 | 0.227 | 0.977 |
| APH activity/nmol·min$^{-1}$. 50 seeds | Year (Y) | 1 | 0.003 | 0.957 |
| | N treatment (N) | 7 | 1.461 | 0.200 |
| | Interaction Y×N | 7 | 0.244 | 0.972 |

**Note:**
HSY, harvested seed yield; SGR, standard germination rate; GE, germination energy; AAGR, accelerated aging germination rate; DE, dehydrogenase; APH, acid phosphoesterase; Df, degrees of freedom.

## Comprehensive analysis of membership function and heat map

The changes in the seed trait indices of *K. melanthera* represented a certain degree of correlation and difference based on the N fertilizer treatments, and a single index cannot fully explain the influence of N fertilizer treatment on plant reproduction and seed production. Consequently, it was necessary to conduct a comprehensive evaluation of multiple indices to determine the optimal amount of N fertilizer application. Based on the values of each seed trait index measured under different N treatments, including indices of six seed yield components, three seed yield, and five seed quality, to calculated the membership function values (Table 5). Higher membership function values represented more distinct improvement of *K. melanthera* seed yield or quality. The results indicated that the optimal deses of N fertilizer treatment was $N_6$ (followed closely by $N_7$) in both 2016 and 2017 (Table 5 and Table S2).

Up-regulation or down-regulation of the indices for seed yield components, seed yield, and seed quality were revealed by the heatmap and hierarchical cluster analysis. In the first trial year, up-regulation of all indices was most obvious for $N_6$, followed by $N_5$ and $N_7$ (Fig. 3A). In addition, $N_5$, $N_6$, $N_7$, and $N_8$ could be classified into one group through hierarchical cluster analysis. Meanwhile, we also noted that germination energy (GE) and acid phosphoesterase (APH) revealed significant up-regulations at $N_3$ and $N_4$, respectively. In the second trial year, up-regulation of most indices were relatively significant for both $N_6$ and $N_7$, and could be classified into one group through cluster analysis (Fig. 3B).

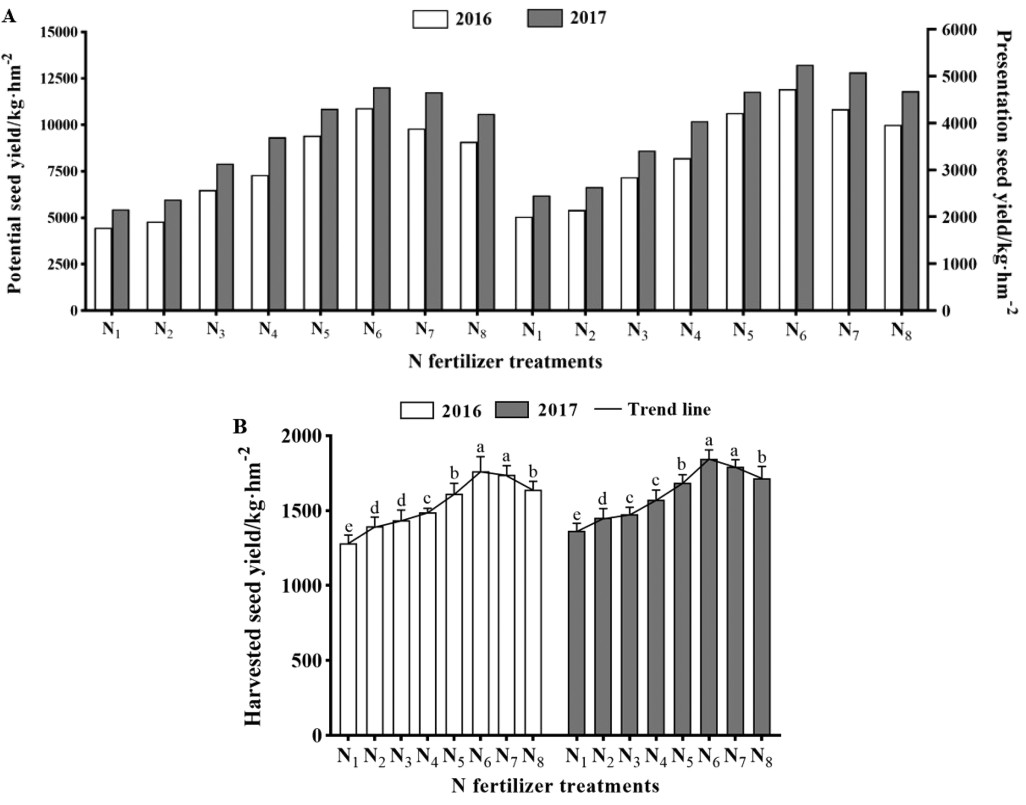

**Figure 2** (A) The seed yield of potential, presentation, and (B) harvested (mean ± standard error, $n = 4$) of *K. melanthera* as influenced by N fertilizer during the trial years of 2016–2017. Different lower-case letters above the column indicate significant differences under different N fertilizer treatments in each trial year (Bonferroni, $P < 0.05$). Vertical bar represents the standard error of mean. Note: $N_1$ – N fertilizer 0 kg·hm$^{-2}$ (Control); $N_2$ – N fertilizer 60 kg·hm$^{-2}$; $N_3$ – N fertilizer 90 kg·hm$^{-2}$; $N_4$ – N fertilizer 120 kg·hm$^{-2}$; $N_5$ – N fertilizer 150 kg·hm$^{-2}$; $N_6$ – N fertilizer 180 kg·hm$^{-2}$; $N_7$ – N fertilizer 210 kg·hm$^{-2}$; $N_8$ – N fertilizer 240 kg·hm$^{-2}$.

Interestingly, accelerated aging germination rate (AAGR) and standard germination rate (SGR) exhibited down-regulation at $N_6$. Therefore, the heat maps obtained the same results as the comprehensive analysis of membership function in both trial years, that is $N_6$ (180 kg·hm$^{-2}$ urea) was the optimal N fertilizer treatment.

# DISCUSSION

## Effect of N on seed yield

The main features of elite varieties of perennial grass are known to increase the yield of aboveground biomass and vegetation coverage of grassland stands thereby meets the needs of grassland husbandry and ecological restoration (*Varadi, Kadar & Duda, 2019*). Numerous investigations of increasing seed yield through different agronomic management practices have been carried out for many perennial grasses in the QTP, *e.g.*, *Elymus sibiricus, E. nutans*, and *Phalaris arundinacea*, but few have probed into the effects of fertilization on seed yield. It is generally believed that a low fertilization rate results in seed yield not reaching the desired goal, but excess fertilizers led to increasing environmental pollution and reducing of seed yield (*Zhang, 2017*). Sufficient nitrogen (N)
**Table 3 Seed quality of *K. melanthera* as influenced by N fertilizer during trial years 2016–2017.**

| Year | N treatment | SGR (%) | GE (%) | AAGR (%) | DE activity (µg/mL) | APH activity (nmol/min. 50 seeds) |
|------|-------------|---------|--------|----------|---------------------|-----------------------------------|
| 2016 | $N_1$ | 86.00 ± 3.92a | 83.50 ± 4.03a | 64.00 ± 2.58a | 15.17 ± 0.59a | 2.39 ± 0.17a |
|      | $N_2$ | 86.50 ± 3.50a | 83.50 ± 4.43a | 64.50 ± 4.19a | 15.45 ± 0.61a | 2.41 ± 0.26a |
|      | $N_3$ | 86.00 ± 3.65a | 84.00 ± 2.94a | 65.00 ± 2.65a | 15.38 ± 0.60a | 2.47 ± 0.24a |
|      | $N_4$ | 86.50 ± 3.50a | 83.50 ± 3.30a | 64.00 ± 3.65a | 15.72 ± 0.97a | 2.52 ± 0.40a |
|      | $N_5$ | 87.00 ± 3.70a | 83.50 ± 4.11a | 65.50 ± 3.59a | 15.88 ± 0.59a | 2.50 ± 0.25a |
|      | $N_6$ | 87.50 ± 4.03a | 84.00 ± 4.08a | 66.00 ± 2.94a | 16.23 ± 1.04a | 2.50 ± 0.20a |
|      | $N_7$ | 86.50 ± 4.65a | 83.50 ± 2.28a | 65.00 ± 3.42a | 15.96 ± 0.76a | 2.48 ± 0.36a |
|      | $N_8$ | 86.50 ± 3.59a | 83.50 ± 3.10a | 66.00 ± 4.55a | 16.12 ± 0.73a | 2.45 ± 0.28a |
| 2017 | $N_1$ | 86.00 ± 9.38a | 83.00 ± 2.38a | 65.00 ± 4.51a | 15.28 ± 0.57a | 2.42 ± 0.18a |
|      | $N_2$ | 86.50 ± 8.23a | 83.50 ± 4.11a | 65.00 ± 3.32a | 15.54 ± 0.58a | 2.40 ± 0.30a |
|      | $N_3$ | 87.00 ± 3.46a | 82.00 ± 3.74a | 66.00 ± 2.83a | 15.85 ± 0.52a | 2.49 ± 0.37a |
|      | $N_4$ | 86.50 ± 7.19a | 82.50 ± 4.03a | 65.00 ± 3.42a | 16.34 ± 0.88a | 2.45 ± 0.29a |
|      | $N_5$ | 87.00 ± 11.60a | 82.50 ± 2.87a | 64.50 ± 3.87a | 16.03 ± 0.79a | 2.50 ± 0.35a |
|      | $N_6$ | 86.50 ± 10.38a | 84.00 ± 3.56a | 65.00 ± 3.30a | 16.49 ± 0.74a | 2.48 ± 0.29a |
|      | $N_7$ | 87.00 ± 8.67a | 83.50 ± 2.63a | 66.00 ± 1.63a | 16.44 ± 0.46a | 2.49 ± 0.38a |
|      | $N_8$ | 86.50 ± 7.72a | 83.50 ± 3.40a | 64.00 ± 2.45a | 15.95 ± 0.67a | 2.47 ± 0.34a |

**Note:**
Measurement results are means ± standard error ($n = 4$). In the same column, standard error of mean followed by the same lower-case letters indicate no-significant differences under different N fertilizer treatments in each trial year base on a Bonferroni multiple comparison test at $P < 0.05$. SGR, standard germination rate; GE, germination energy; AAGR, accelerated aging germination rate; DE, dehydrogenase; APH, acid phosphoesterase; $N_1$ – N fertilizer 0 kg·hm$^{-2}$ (Control); $N_2$ – N fertilizer 60 kg·hm$^{-2}$; $N_3$ – N fertilizer 90 kg·hm$^{-2}$; $N_4$ – N fertilizer 120 kg·hm$^{-2}$; $N_5$ – N fertilizer 150 kg·hm$^{-2}$; $N_6$ – N fertilizer 180 kg·hm$^{-2}$; $N_7$ – N fertilizer 210 kg·hm$^{-2}$; $N_8$ – N fertilizer 240 kg·hm$^{-2}$.

**Table 4 Path analysis of seed yield to yield components and N application.**

| Source | NFTs/m² | NSPs per fertile tillers | NFLs per spike | NFFLs per spike | TSW/g | Seed yield |
|--------|---------|--------------------------|----------------|------------------|-------|------------|
| N treatment | 0.950* | 0.538 | 0.683 | 0.739 | 0.921* | 0.909* |
| Seed yield | 0.944* | 0.806 | 0.818 | 0.899* | 0.912* | – |

**Note:**
The value in the table is the correlation coefficient "r". Statistical significance was determined as * $P < 0.05$. NFTs, number of fertile tillers; NSPs, number of spikes; NFLs, number of florets; NFFLs, number of fertile florets; TSW, 1,000-seed weight.

in the soil is the basis for high seed yield. Seed yield is significantly positively correlated with N fertilizer dose within a specific range, even if no other fertilizer is applied (*Wang et al., 2005a*). Most researchers choose N fertilizer to increase seed yield as it is the most consumable component during plant growth and seed maturation (*Li et al., 2018b*). In addition, N fertilizer can not only improve the developmental structure of plant roots and absorption of nutrients, but also the formation and size of grains, which can directly determine the seed yield, were promoted during later stages of plant growth (*Liu et al., 2020*). Therefore, compared with other fertilizers, N fertilizer is one of the most effective measures to improve seed yield (*Oliveira et al., 2007*). By combining seed yield indices (harvested seed yield (HSY), potential seed yield (PSY) and presentation seed yield (PRSY)) under different N application doses ($N_1$–$N_6$), we propose an optimal rate of N fertilizer application ($N_6$) affected seed yield in *K. melanthera* (see in Figs. 2A and 2B). Similar results could be observed with an increase in the number of seeds or the weight of a

Yuan et al.
2022
10.7717/peerj.14101

...

**Table 5 Membership function analysis based on multiple indicators for the effects of N fertilizer treatments on *K. melanthera* during trial year 2016.**

| Parameter | N fertilizer treatments | | | | | | | |
|---|---|---|---|---|---|---|---|---|
| | $N_1$ | $N_2$ | $N_3$ | $N_4$ | $N_5$ | $N_6$ | $N_7$ | $N_8$ |
| NTs/m$^2$ | 0.00 | 0.05 | 0.21 | 0.43 | 0.71 | 0.89 | 1.00 | 0.99 |
| NFTs/m$^2$ | 0.00 | 0.02 | 0.33 | 0.51 | 0.85 | 1.00 | 0.97 | 0.95 |
| TSW/g | 0.00 | 0.30 | 0.36 | 0.42 | 0.88 | 1.00 | 0.97 | 0.95 |
| NSPs per fertile tillers | 0.00 | 0.20 | 0.75 | 0.70 | 0.90 | 1.00 | 0.60 | 0.35 |
| NFLs per spike | 0.00 | 0.00 | 0.23 | 0.38 | 0.54 | 1.00 | 0.62 | 0.31 |
| NFFLs per spike | 0.00 | 0.00 | 0.25 | 0.50 | 0.75 | 1.00 | 0.75 | 0.25 |
| PSY | 0.00 | 0.05 | 0.32 | 0.44 | 0.77 | 1.00 | 0.83 | 0.72 |
| PRSY | 0.00 | 0.05 | 0.31 | 0.46 | 0.81 | 1.00 | 0.84 | 0.72 |
| HSY/kg·hm$^{-2}$ | 0.00 | 0.24 | 0.32 | 0.43 | 0.69 | 1.00 | 0.95 | 0.74 |
| SGR/% | 0.00 | 0.33 | 0.00 | 0.33 | 0.67 | 1.00 | 0.33 | 0.33 |
| GE/% | 0.00 | 0.00 | 1.00 | 0.00 | 0.00 | 1.00 | 0.00 | 0.00 |
| AAGR/% | 0.00 | 0.25 | 0.50 | 0.00 | 0.75 | 1.00 | 0.50 | 1.00 |
| DE activity/μg·mL$^{-1}$ | 0.00 | 0.26 | 0.20 | 0.52 | 0.67 | 1.00 | 0.75 | 0.90 |
| APH activity/nmol·min$^{-1}$. 50 seeds | 0.00 | 0.15 | 0.62 | 1.00 | 0.85 | 0.85 | 0.69 | 0.46 |
| Rank | 8 | 7 | 6 | 5 | 3 | 1 | 2 | 4 |

Note:
NTs, number of tillers; NFTs, number of fertile tillers; NSPs, number of spikes; NFLs, number of florets; NFFLs, number of fertile florets; TSW, 1,000 seeds weight; PSY, potential seed yield; PRSY, presentation seed yield; HSY, harvested seed yield; SGR, standard germination rate; GE, germination energy; AAGR, accelerated aging germination rate; DE, dehydrogenase; APH, acid phosphoesterase; $N_1$ – N fertilizer 0 kg·hm$^{-2}$ (Control); $N_2$ – N fertilizer 60 kg·hm$^{-2}$; $N_3$ – N fertilizer 90 kg·hm$^{-2}$; $N_4$ – N fertilizer 120 kg·hm$^{-2}$; $N_5$ – N fertilizer 150 kg·hm$^{-2}$; $N_6$ – N fertilizer 180 kg·hm$^{-2}$; $N_7$ – N fertilizer 210 kg·hm$^{-2}$; $N_8$ – N fertilizer 240 kg·hm$^{-2}$.

single seed (*Martiniello, 1998*). The reason may be that the N accumulated in the vegetative organs is transported and redistributed to the grains, and thus enhancing seed yield due to fuller grains (*Yi et al., 2020*). As N fertilization continues to rise, seed yield indices were significantly decreased from $N_7$ to $N_8$. This may have been caused by excessive growth of competing plants encouraged by excessive N fertilizer application, or by dehydration of plant roots (*Aly, 1993*). It happens that there is a similar case, a study of a wildrye *Elymus nutans* in Tibet, which showed a peak yield of 2016 kg·hm$^{-2}$ at a N fertilizer dose of 250 kg·hm$^{-2}$, and a trend of falling continuously for HSY when the N fertilizer dose was continued increased (*Song et al., 2008*). Therefore, we should manage proper N fertilizer rate to optimize N-uptake in seed production of *K. melanthera*.

## Effect of N on seed yield components

For most grass species, seed yield always depends on the multiplicative effect of the yield components (number of tillers (NTs), number of fertile tillers (NFTs), number of spikes (NSPs), number of florets (NFLs), number of fertile florets (NFFLs), and 1,000-seed weight (TSW)) (*Dewitt et al., 2021*). A large number of studies point out that N fertilizer can significantly improve the seed yield components. *Song et al. (2008)* found that N fertilizer could significantly raise the NFTs of *E. nutans* to improve the number of seeds per plant and thus increase seed yield. *Xiao (1984)* obtained a similar result, namely that the NFTs

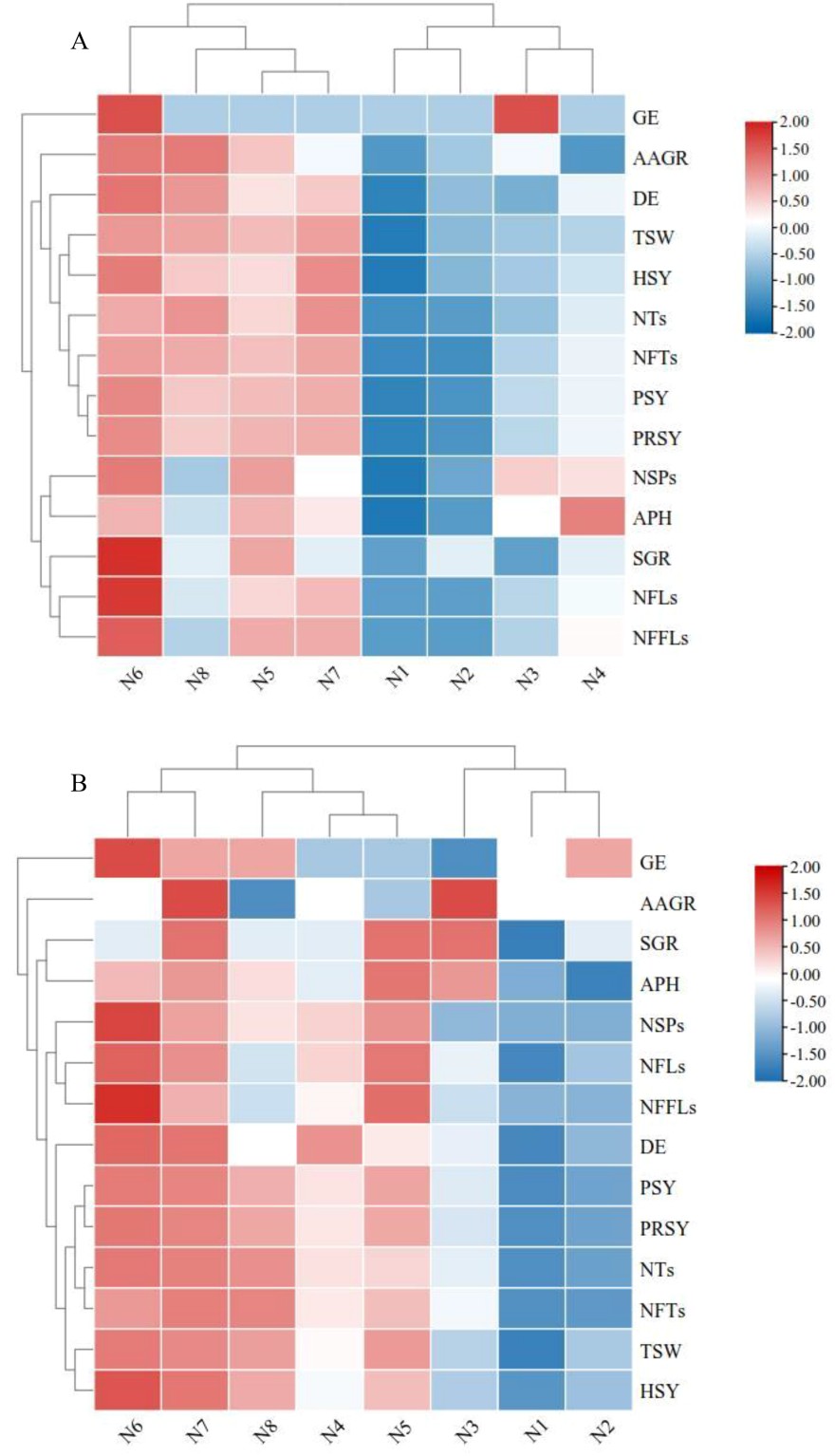

**Figure 3 Heat maps. Log₂-fold change of *K. melanthera* seed yield and quality under N fertilizer treatments during trial year 2016 (A) and 2017 (B).** Red is up-regulated, blue is down-regulated. Note: NTs, number of tillers; NFTs, number of fertile tillers; NSPs, number of spikes; NFLs, number of florets; NFFLs, number of fertile florets; TSW, 1,000-seed weight; PSY, potential seed yield; PRSY,

**Figure 3** (continued)
presentation seed yield; HSY, harvested seed yield; SGR, standard germination rate; GE, germination energy; AAGR, accelerated aging germination rate; DE, dehydrogenase; APH, acid phosphoesterase; $N_1$ – N fertilizer 0 kg·hm$^{-2}$ (Control); $N_2$ – N fertilizer 60 kg·hm$^{-2}$; $N_3$ – N fertilizer 90 kg·hm$^{-2}$; $N_4$ – N fertilizer 120 kg·hm$^{-2}$; $N_5$ – N fertilizer 150 kg·hm$^{-2}$; $N_6$ – N fertilizer 180 kg·hm$^{-2}$; $N_7$ – N fertilizer 210 kg·hm$^{-2}$; $N_8$ – N fertilizer 240 kg·hm$^{-2}$.                         

were higher with abundant N supply. For seed production of *Poa pratensis*, *Zhou & Lu (2008)* showed that N fertilizer significantly boosted NFTs and spike weight, thus enhancing seed yield. Investigation of the seed yield of tall fescue accounted for the fact that the amount of N fertilizer application is proportional to NFTs, NFLs and NFFLs per spike within a particular range (*Ma et al., 2003*). In addition, similar findings have been reported with regard to the seed yield of *Lolium perenne*, N fertilizer increased NSPs per fertile tiller, NFFLs per spike, and TSW (*Hides, Kute & Marshall, 2010*; *Elgersma, Nijs & Eeuwijk, 1989*). In this study, seed yield (r = 0.909, $P < 0.05$), NFTs/m$^2$ (r = 0.950, $P < 0.05$) and TSW (r = 0.921, $P < 0.05$) were dramatically positively correlated with N fertilizer treatment by means of path analysis (Fig. 1 and Table 4). This result indicates that N treatment was one of the most effective approaches to improving *K. melanthera* production by enhancing NFTs and TSW. Furthermore, we also found that seed yield was non-significantly correlated with NSPs per fertile tiller (r = 0.806, $P > 0.05$) and NFLs per spike (r = 0.818, $P > 0.05$). These findings are not entirely consistent with other studies. *Wang et al. (2005a)* researched the influence of five yield components on the seed yield of *Dactylis glomerata*, and determined the following order of influence: NFTs/m$^2$ > NFLs/spike > NFFLs/spike > TSW > NFLs/fertile tiller. However, a different order was obtained for *K. melanthera*: NFTs/m$^2$ > TSW > NFFLs/spike > NFLs/spike > NFLs/fertile tiller. The reasons for the discrepancies may be the difference in sampling regions and studied species.

## Effect of N on seed quality

The seed quality can be reflected by the standard germination rate (SGR), germination energy (GE), and accelerated aging germination rate (AAGR). The higher the values are for these, the stronger the seed vigor (*Lin et al., 2020*). It is worth noting that when AAGR is compared to SGR, the aging-accelerated conditions can explain the resistance to adversity that supplies a more reliable reference for seed growth in specific environments such as the QTP (*Li & Mao, 2013*; *Chen et al., 2017*). The number of SGR was inferred under suitable environmental conditions and cannot guarantee actual germination rate in the field and survivability under stress, therefore, it is always in synergy with AAGR to evaluate potential viability of seed (*Mao, Hou & Wang, 2016*). Dehydrogenase (DE) and acid phosphoesterase (APH) activity are often used as crucial indices to test vigor and utilization value of forage seed, because these two indices more accurately reflect the quality and stress resistance of seeds in different stages of growth than seed germination count (*Li & Mao, 2013*; *Chen et al., 2017*). Currently, a large number of researchers have shown that N fertilizer is a key factor in increasing seed yield of perennial herb, but the effect of N on pasture seed quality varied with the different species, growing conditions,

harvesting methods, and/or harvesting time (*Liu, 2020*). The SGR and GE of *Medicago sativa* (*Wang, 2005b*) and *Festuca elata* (*Cheng et al., 2003*) were significantly increased by N fertilizer application, as well as APH activity of *Festuca elata* were also significantly upward. The reason may be that N fertilizer can promote the transfer of plant metabolites to reproductive organs and consequently increase plumpness and 1,000-seed weight of plant seed. However, the study of *Qiao & Han (2010)* demonstrated that N application had no significant effect on seed quality of *E. nutans*, a native perennial grass in QTP region. The results of this study were in agreement with those results of present study, which the seed quality indices (GR, GE, AAGR, DE and APH activity) of *K. melanthera* were not significantly affected by N fertilizer in either 2016 or 2017 (Table 2).

## CONCLUSION

*K. melanthera* is a beneficial multi-purpose perennial grass in the eastern QTP region. It is commonly used in the treatment of degraded and sandy grassland, as well as for artificial planting to supply forage for livestock. However, the limited supply and poor quality of seeds have restricted its popularization and utilization in QTP. In this work, a two-year trial in eastern QTP was implemented to evaluate the effects of seed yield and quality of *K. melanthera* under different N fertilizer levels. The results showed that the optimal N application rate was 180 kg·hm$^{-2}$ during trial years 2016–2017, and most of seed yield indicators reached the respective highest values at this N level, including the harvested seed yield (HSY), potential seed yield (PSY), presentation seed yield (PRSY), number of tillers (NTs), number of fertile tillers (NFTs), and 1,000-seed weight (TSW). The present findings provide a practical technical groundwork for improving seed productivity of *K. melanthera* in eastern QTP.

### Funding
This work was supported by the Sichuan Provincial Key Science and Technology Project (2019YFN0170, 2022YFQ0076). The funders had no role in study design, data collection and analysis, decision to publish, or preparation of the manuscript.

### Grant Disclosures
The following grant information was disclosed by the authors:
Sichuan Provincial Key Science and Technology Project: 2019YFN0170, 2022YFQ0076.

### Competing Interests
The authors declare that they have no competing interests.

### Author Contributions
- Shuai Yuan conceived and designed the experiments, performed the experiments, analyzed the data, prepared figures and/or tables, authored or reviewed drafts of the article, and approved the final draft.

- Yao Ling conceived and designed the experiments, performed the experiments, analyzed the data, prepared figures and/or tables, authored or reviewed drafts of the article, and approved the final draft.
- Yi Xiong conceived and designed the experiments, performed the experiments, analyzed the data, prepared figures and/or tables, authored or reviewed drafts of the article, and approved the final draft.
- Chenglin Zhang conceived and designed the experiments, performed the experiments, analyzed the data, prepared figures and/or tables, authored or reviewed drafts of the article, and approved the final draft.
- Lina Sha performed the experiments, authored or reviewed drafts of the article, and approved the final draft.
- Minghong You performed the experiments, authored or reviewed drafts of the article, and approved the final draft.
- Xiong Lei performed the experiments, authored or reviewed drafts of the article, and approved the final draft.
- Shiqie Bai performed the experiments, analyzed the data, prepared figures and/or tables, authored or reviewed drafts of the article, and approved the final draft.
- Xiao Ma conceived and designed the experiments, performed the experiments, analyzed the data, prepared figures and/or tables, authored or reviewed drafts of the article, and approved the final draft.

## Data Availability

The raw measurements are available in the Supplemental Files.

## Supplemental Information

Supplemental information for this article can be found online at http://dx.doi.org/10.7717/peerj.14101#supplemental-information.

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
