# Peer review of "Effect of nitrogen fertilizer on seed yield and quality of Kengyilia melanthera (Triticeae, Poaceae)"

_PeerJ, doi:10.7717/peerj.14101_

## Round 0.1 · original submission · Major Revisions

Please modify according to the comments of reviewers.

Reviewer 1 ·

Basic reporting

The English used throughout the manuscript is clear, unambiguous, and professional.
I would suggest some minor changes:
1. There are a lot of abbreviated forms in the manuscript which makes it difficult to read, I would suggest either the full form of some or include a different section mentioning the full forms of all the abbreviated terms.

Experimental design

1. Enlarge figure 1. labels are difficult to read.
2. In the section of the Experimental site, there is no need to provide the results of soil, instead it should be included in the result section. (lines 98 to 101).

Validity of the findings

1. The year of studies are 5-7 years ago, and the climate and other parameters are significantly changing every year. Justify the relevance of this work in today's environment.
2. Explain the reason behind using hierarchical cluster analysis.

Reviewer 2 ·

Basic reporting

Please see the attached file

Experimental design

Please see the attached file

Validity of the findings

Please see the attached file

Annotated reviews are not available for download in order to protect the identity of reviewers who chose to remain anonymous.

Reviewer 3 ·

Basic reporting

The Qinghai-Tibet Plateau (QTP), the so-called third pole of the world, is an important eco-region. Widely distributed in the alpine sandy grassland in east Qinghai-Tibet Plateau (QTP), Kengyilia melanthera is considered as an ideal pioneer grass for the restoration of degraded and desertification grassland in the region. Under the special ecological and climatic conditions in the northwest Sichuan plateau located in east QTP, it is of great significance to optimize the amount of nitrogen fertilizer for the seed production of this species. This study provided a certain practical suggestion for the improvement of seed production of K. melanthera in the northwest Sichuan plateau. Whereas the topic is interesting, I think this research falls short in several ways.
1. The description of the preface of the paper is confusing and has little relevance to the content shown in this study. It is recommended to add more research papers on the Effect of nitrogen fertilizer on seed yield and quality
2. Using herbicide spray in the description of the experimental site paragraph, will this affect the germination and growth of the experimental seeds?
3. What is the main reference for the 8 different nitrogen application rates, and it is recommended to explain, why set up 8 gradients?
4. Is it necessary to keep Table3?
5. line101: PH 6.02 is recommended to be written as pH 6.02.
6. The discussion of the thesis is not in-depth enough, and it is necessary to discuss in depth with the content of the results to condense the research conclusions of the thesis

Experimental design

no comment

Validity of the findings

no comment

Additional comments

no comment

---

## Round 0.2 · Minor Revisions

Please revise according to the comments of reviewers.

Reviewer 2 ·

Basic reporting

After revision, the background, topic, results of the MS have been well described, data which were listed in tables and figures are sufficient to support the viewpoints of authors.

Experimental design

Authors have given more clear descriptions on the details of experimental design.

Validity of the findings

the novelty of this study still needs further description. "However, studies on the effect of N application rate of seed quality and quantity of K. melanthera have not been reported up to now." is not enough as a scientific issue for a journal mainly publishing fundamental researches.
Conclusion in this version is not well written, it looks like a simple summary on the meaning of this studied while important results are not well showen.

---

## Round 0.3 · accepted · Accept

The article has been revised according to the comments of reviewers and can be accepted.